# Adopting Andersen's behavior model to identify factors influencing maternal healthcare service utilization in Bangladesh

Md. Ruhul Kabir[1,2]*

**1** School of Communication, Hong Kong Baptist University, Kowloon Tong, Hong Kong, **2** Department of Food Technology & Nutrition Science, Noakhali Science & Technology University, Noakhali, Bangladesh

* 20481713@life.hkbu.edu.hk, ruhul109@gmail.com

## Abstract

### Background

Equitable maternal healthcare service access and it's optimum utilization remains a challenge for many developing countries like Bangladesh, and different predisposing, enabling, and need-based factors affect the level of maternal healthcare use. The evidently poor maternal healthcare service utilization and disparities among groups in Bangladesh are concerning considering its effect on maternal health outcomes. The study aimed to identify the factors that influence maternal healthcare service (MHS) utilization in Bangladesh by adopting Andersen's behavior model of health service use as the theoretical framework.

### Methods

The 2017–18 Bangladesh Demographic Health Survey (2017–18 BDHS) data were used which is nationally representative. The survey study used two-stage stratified sampling to select study households, and data were collected through face-to-face interviews. The desirable, moderate, and undesirable maternal health service (MHS) package was developed based on antenatal, and delivery care services use during pregnancy and childbirth. Multinomial logistic regression and discriminant analysis were performed to analyze the factors that affect MHS use.

### Results

Out of 5,011 ever-married women, only 31.2% of women utilized the desirable level of MHS. The likelihood of using the desirable level of MHS package, relative to the undesirable category, was 9.38 times (OR: 9.38, 95% CI: 4.30–20.44) higher for women with a higher level of education compared to illiterate women, and the same trend was noticed for husband's education. The wealth index had the highest standardized function coefficients (Beta coefficient: 0.49) in discriminatory function. Women with the richest wealth index were more than 23 times (OR: 23.27, 95% CI: 12.69–42.68) likely to have utilized desirable MHS than their poorest counterparts. The likelihood of service uses also varied according to the child's birth order, administrative regions, and area of residence (rural vs. urban).

**Data Availability Statement:** All relevant data are within the paper and its Supporting Information files.

**Funding:** The author(s) received no specific funding for this work.

**Competing interests:** The authors have declared that no competing interests exist.

## Conclusions

Policies and interventions directed towards poverty reduction, universal education, and diminishing geographical disparities of healthcare access might influence the desirable use of maternal healthcare services in Bangladesh.

## Introduction

Inadequate utilization of maternal healthcare services continues to be a major concern for many developing regions in the world [1, 2], despite being a central element in their development agendas. The major burden of maternal mortality and morbidities falls on low- and middle-income countries (LMICs) for a number of different reasons: inequitable access, affordability, availability, and poor quality are at the top of the list [3]. Antenatal (ANC) and delivery care, the key components of safe motherhood, play a pivotal role in tackling preventable maternal deaths [4, 5]. Antenatal care, care before birth, provides essential health services to pregnant women, including regular physical examination and essential advice to prepare women for safe delivery. The ANC component aids women in detecting pregnancy problems, and its usefulness is dependent primarily on receiving care from healthcare providers during the pregnancy [6]. Furthermore, ANC has a strong influence on ensuring skilled birth attendance during delivery and boosting facility delivery rates [7, 8]. Delivering at a health facility by skilled health professionals can produce better health outcomes [9]. Postnatal care after delivery can also play an important role in ensuring the safe return of mothers to home [10]. These reproductive health services are essential for better maternal health outcomes and lowering maternal mortality rates [11].

Bangladesh, one of the South Asian countries, experiences very high maternal mortality (currently 173 deaths per 100,000), and the level of healthcare utilization by mothers are below the expected level [12]. Only 47% of the women make recommended number of ANC visits. Skilled health professionals assisted only 53% of the deliveries, and half of the deliveries do not take place at health facilities. The utilization of these health services varies according to many factors, and the widespread inequitable utilization of health services was evident [13]. One study conducted in Bangladesh reported unfavorable social determinants as one of the important reasons for low levels of reproductive health services use in urban women. The study also found a wide spectrum of inequalities across the wealth status of the women [14]. Another study reported that cultural practice, cost of delivery, poor quality of service, presence of male delivery assistant, and lack of knowledge and preparation were among the important reasons for not using institutional delivery services, and there was the presence of an unequal distribution of health service use among women of the different socioeconomic spectrum [15].

Despite the adoption and implementation of different programs directed to improve maternal health and improve access by the government, inequity in the utilization of services is pervasive, and women from the lower socioeconomic spectrum are the victim of it. For enhancing the provision of equitable maternal health service access, identifying factors and addressing them should be the top priority to tackle the unusual rate of maternal mortality in Bangladesh [16].

Therefore, to identify factors that influence maternal health service, this present study developed an indicator variable, namely the maternal healthcare service (MHS) package, combining four important maternal health service variables. The four variables that comprise the MHS package include the number of ANC visits, ANC assistance, place of delivery, and

delivery assistance. To identify the predictors, Andersen's behavioral model of health service use (HBM) was used to guide and select predictors of health service use. The theoretical implications of Andersen's model assist in targeting important factors that have been widely used in health service research [17]. Therefore, the study's objective was to investigate the likelihood of utilizing the MHS package and the factors that contribute most to the inequitable access and utilization of the MHS package. The study findings can contribute to developing policies and interventions to facilitate service access and use.

## Methods

### Conceptual framework: Andersen's behavioral model (HBM) of health service use

The study was guided by and adapted from Andersen's Behavioral Model of health service use which argues healthcare service use as a ramification of three important component functions. The HBM model attempted to explain the reason (why and how) of healthcare service use, the predisposing factors that influence acute healthcare service use, enabling factors that facilitate or barriers use, and the subsequent perceived needs to use healthcare. The model enables to appraise measures of healthcare access, equitability, effectivity, and efficiency and understand the environmental influence on it. The model argues certain external environmental and individual characteristics (predisposition, enabling, and need element) may guide the disposition of health behavior to use health service, which later influences health outcomes. The predisposing factors, which include socio-demographic characteristics, socio-structural and behavioral factors, may facilitate by enabling personal/family/community resources, and the perceived and evaluated need of healthcare influences access and healthcare service use decision [17–19].

The study hypothesized that the selected predisposing, enabling, and need factors significantly predict the outcome variable based on the conceptual model. The model later expanded on to include race and ethnicity and the importance of health belief constructs in health service use [20]. The Andersen's model also discussed the importance of mutability for using the HBM to promote equitable access to healthcare. For significant behavioral change to happen, a variable requires to be mutable enough (that might bring change) so that policy changes can influence behavioral adoption. Demographic and social structure variables that represent predisposing factors might possess low mutability. Variables like gender, ethnicity, or age cannot be altered for changing health utilization. Changing educational or occupational structures might not be possible to change in the short term as well [17]. According to HBM, some of the enabling factors (e.g., health insurance) may have strong mutable characteristics and dramatically impact health service use. Need factors also have low mutability; however, different education programs or financial incentives can influence people's perceived needs. Based on the mutability characteristics, it appeared that enabling factors have high mutability, which can be planned to change to affect healthcare use [17]. Therefore, the study also hypothesized that enabling factors might have a greater predictive influence on health care use. The figures below illustrate the adapted HBM model of health service use (Fig 1), including the outcome and predictor variables and their mutability properties (Fig 2).

### Data source and sampling

The study used the 2017–18 Bangladesh Demographic Health Survey (2017–18 BDHS) data, which provided a nationally representative sample. The study was conducted on households of ever-married women aged 15–49 years to report updated information on the demographic and health status of the community. The eighth national survey used a list of enumeration

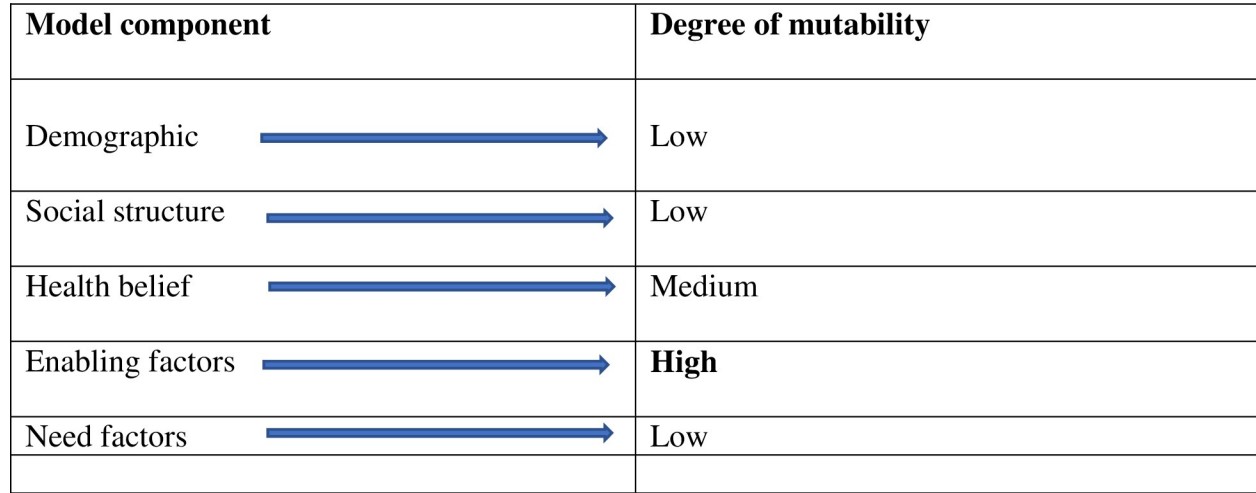

**Fig 1. Andersen's behavior model (adapted) on determining the utilization of maternal healthcare service (MHS) package utilization [17].**

areas (EAs) as a sample frame consisting of about 120 households gleaned from a double-staged stratified sample of households. Data were collected through face-to-face interviews by well-trained field staff. The survey included questions (on women questionnaire) that involve

| Model component | Degree of mutability |
|---|---|
| Demographic | Low |
| Social structure | Low |
| Health belief | Medium |
| Enabling factors | **High** |
| Need factors | Low |
|  |  |

**Fig 2. Concept of mutability adapted from Andersen's healthcare service utilization model [17].**

information on women's background characteristics (age, education, religion, etc.), reproductive and family planning history, maternal medical care, breastfeeding practices, etc. The 2017–18 BDHS data were collected by maintaining all the ethical standards. The study protocol was approved by the National Ethics Review Committee, Bangladesh Ministry of Health and Family Welfare [13]. To obtain data access, a brief description of the study plan was submitted to the Measure DHS website (https://dhsprogram.com/), and upon approval of the study, the data was extracted. There are several formats of data available there, and the SPSS format was chosen for this study. After careful observation of data on outcome and predictor variables, a total of 5,011 ever-married women aged 15–49 years who delivered at least one child three years prior to the survey were included in the present study. If women had more than a child in the last three years, then information about the latest pregnancy was considered to be included. The analysis reported in this study was based on these samples for whom information was available for most of the variables considered for the study.

## Outcome variable

The maternal healthcare service (MHS) package utilization was the outcome variable, an indicator variable generated from four variables: number of ANC visits, ANC provider, place of delivery, and delivery assistant. These four variables were chosen based on their importance on maternal health, and the level of maternal healthcare service utilization was considered depending on the probability of the women utilizing a particular package of MHS service. The idea of the MHS package as an outcome variable was adopted from the study conducted by Rutaremwa et al. (2015) by using the 2011 Uganda Demographic Health Survey Data (UDHS) [21]. Initially, postnatal care (PNC) service utilization was also considered to be included in the package; however, it appeared that most of the women who utilized delivery care were able to receive PNC care. Therefore, adding PNC care to the MHS package would not have added any value. The categorization of the outcome variable has been provided in Table 1.

## Predictor variables

As mentioned in the behavioral model of health service use, the study considered a wide range of predictors to be included in the model. The categorization of the predictors has been provided in Table 1 under three major elements of the healthcare service utilization function.

## Data management and analysis

Data editing, coding, and analysis were done by using statistical software SPSS version 25.0. Descriptive statistics using frequency distribution and percentages were presented for background characteristics. The association of predictors with outcome variables was examined by applying Pearson's chi-square test. The predictors which significantly ($p < 0.05$) affected outcome variables in bivariate analysis were included in the multivariate multinomial logistic regression analysis (MLRM) to determine factors that best predict MHS package utilization. According to the SAGE dictionary, "The multinomial logistic regression model allows each category of an unordered outcome variable to be compared to an arbitrary reference category (usually the last category), providing a number of logit models. A multinomial logistic regression model equation is as follows [22]:

$$\text{Log } P_r(Y = j)/P_r(Y = j') = \alpha + \beta_1 X_1 + \beta_2 X_2 + \ldots + \beta_k X_k$$

Where *j* is the identified response category and *j'* is the reference response category. The multinomial model generates j-1 sets of parameter estimates, one for each category relative to the reference category, to explain the relationship between the DV and the IVs".

**Table 1. Variable's categorization and leveling.**

| Variables | Description | Categories |
|---|---|---|
| **Outcome variable:** Maternal healthcare service (MHS) package utilization [21] | Maternal healthcare service (MHS) package utilization as indicator outcome variable generated from the following variables:<br>• Number of antenatal care visits<br>• ANC assistance<br>• Place of delivery<br>• Delivery assistance | 1. Desirable/Ideal category: Women who attended 4 or more ANC visits, ANC assisted by trained health professionals, delivered in a health facility, and delivery assisted by trained health professionals |
| | | 2. Moderate category: Women who have not meet all the criteria of the desirable category but met at least one of the criteria of ANC or delivery care |
| | | 3. Undesirable category: Women who did not attend 4 or more ANC visits, did not assist by health professionals, did not deliver at a health facility, and did not assist by health professionals during delivery |
| **Predictors (Predisposing factors)** | | |
| Women's age | Age of the women | Categorized into three levels: 15–19 years, 20–29 years, 30 or more years |
| Marital status | Current marital status | Categorized into two levels: Married, Divorced/widowed/separated |
| Currently working | Current working status | Categorized into two levels: Currently working, Not working |
| Mother's education | Mother's level of education | Categorized into four levels: Illiterate/no education, Primary school education (Grade 5), Secondary school education (Grade 10), Higher education |
| Religion | The religion of the women | Categorized into two levels: Islam, Others |
| Husband's education | Husband's level of education | Categorized into four levels: Illiterate/no education, Primary school education (Grade 5), Secondary school education (Grade 10), Higher education |
| Husband's occupation | Occupation of the husband | Categorized into five levels: Skilled manual, agricultural, sales/service, professional/technical, others |
| Birth order of the child | Birth order of the latest child | Categorized into three levels: 1–2, 3–4, 5 or more |
| Women's autonomy | Women's autonomy on decisions over their own health. The variables included as a proxy of health belief construct on MHS package use | Categorized into three levels: Wife alone, Husband/wife together, Husband alone/others |
| Area of residence | Area of residence | Categorized into two levels: Rural, urban |
| Division | Administrative region of residence | Categorized into eight levels: Barisal, Chattogram, Dhaka, Khulna, Mymensingh, Rajshahi, Rangpur, Sylhet |
| **Predictors (Enabling factors)** | | |
| Wealth index | Household wealth index was developed using principal component analysis on household assets which includes ownership of durable goods and dwelling characteristics. | Categorized into five levels: Poorest, poorer, middle, richer, richest |
| Health insurance | Health insurance availability | Categorized into two levels: Yes, no |
| **Predictors (Need factors)** | | |
| Visited community clinic | A proxy indicator of need factor generated from the following variables:<br>• Visited community clinics in the last 6 months<br>• Visited temporary clinics in the last 3 months<br>• Visited by family welfare officers in the last 6 months | Categorized into two levels: Visited (yes), Not visited (No) |

In the study, the outcome variable MHS package has three unordered categories (desirable, moderate, and undesirable); the undesirable category was chosen to be the reference category. So, the regression model generates the association of outcome variables with the predictor variables delineating the propensity of a woman using desirable or moderate MHS package relative to undesirable MHS package. The adequacy of the model fit was verified using the likelihood ratio test of goodness of fit, and multicollinearity among predictors was also checked by variance inflation factor values (VIF<2.0). Adjusted odds ratios with 95%

confidence interval (95% CI) values were presented to estimate the level of association with the MHS package utilization. The association was considered significant at p<0.05.

The discriminant analysis was conducted to gauge the mutability characteristics of the HBM model, which characterized enabling factor's high mutability. The discriminant analysis, a multivariate technique, produces discriminant function based on their linear combinations of the predictors that can distinguish or separate the groups. Therefore, by discriminant analysis, it is possible to find out the potential influence of each variable in separating the group variable under study [23]. The purpose of using discriminant analysis in this study was to assess whether enabling factors have more discriminatory properties on the MSH package utilization than other elements of the healthcare function of the HBM model. The only enabling factor in this study considered was the household wealth index since only 0.2% of people were found to be under health insurance and hence, excluded from the study to be included in the final model. Therefore, if the wealth index as an enabling factor has more discriminatory power on the MHS package, it might coincide with the high mutability power interpretation described in the HBM model. It might be interesting to explore the effect of the enabling factor on the model, which is assumed to have high mutability. However, to do the analysis, the outcome variable was categorized into two levels (Desirable and undesirable) to avoid analysis and interpretation complexities of the three levels. Generally, discriminant analysis is performed for a continuous variable with interval data; however, analysis can also be performed with categorical variables, which use indicator (or dummy) variables [24]. The Box's M test of overall model fit provides P-value <0.001, which might indicate not perfect model fit; however, the large data set might be the reason for it and should not be a major issue on the overall fit of the model.

## Results

### Descriptive statistics of study variables

Table 2 provides information on the percentage and frequency distribution of study variables. More than half of the women (61.8) were in the 20–29 years of age category, and almost all (98.7) were married. More than 90% of women were Muslims, and around 37% of them were currently working. The completion of secondary or higher education rate for women and their husband was around 65% and 51%, respectively. Around two-thirds of the husbands were involved in manual skilled or sales and service-related jobs, and 41.8% of the household were either poor or poorer in the household wealth index variable. The Table 1 also reports that the majority (around 65%) of the women lived in rural areas, and 70.9% had their 1st or 2nd child. Every one in four women was not able to make their decision over their own health. More than 65% of the women reported making health decisions with proper consultation with their husbands. Around 41.8% of the women did not visit any community clinics, temporary clinics, or visited by any family planning workers, and there is very little evidence of health insurance existence. Only 31.2% of women utilized desirable or ideal maternal and health service package.

Table 3 describes a bivariate relationship between MCHS package utilization and background variables, and all the variables except women's autonomy were significant predictors of the outcome variable (p<0.05). The table reveals that the utilization of desirable or ideal MHS package use improved with the improved level of education, both for women and their husbands (chi-square: 771.12; p<0.001). Around 60.0% of the women who had higher education utilized desired MHS package, where only 8.0% of the women who had utilized the desirable MHS package were illiterate. Household wealth was also found to have a statistically significant (Chi-square: 1014.06, p<0.001) effect on MHS utilization, and there was a huge

**Table 2. Descriptive statistics according to background characteristics.**

| Characteristics and categories | Number (n) | Percent (%) |
|---|---|---|
| Mother's age group | | |
| 15–19 years | 869 | 17.3 |
| 20–29 years | 3082 | 61.5 |
| 30–49 years | 1060 | 21.2 |
| Marital status | | |
| Married | 4945 | 98.7 |
| Divorced/widowed/separated | 66 | 1.3 |
| Religion | | |
| Islam | 4588 | 91.6 |
| Others | 423 | 8.4 |
| Mother's highest level of education | | |
| Illiterate/No education | 312 | 6.2 |
| Primary School (Grade 5) | 1391 | 27.8 |
| Secondary school | 2402 | 47.9 |
| Higher | 906 | 18.1 |
| Mother's current working status | | |
| Currently working | 1880 | 37.5 |
| Not working | 3131 | 62.5 |
| Husband's highest level of education | | |
| Illiterate/No education | 678 | 13.7 |
| Primary School (Grade 5) | 1657 | 33.6 |
| Secondary school | 1635 | 33.2 |
| Higher | 962 | 19.5 |
| Husband's occupation | | |
| Skilled manual | 1920 | 38.3 |
| Agricultural | 920 | 18.4 |
| Sales/service | 1579 | 31.5 |
| Professional/Technical | 460 | 9.2 |
| Others | 132 | 2.6 |
| Household wealth (quintile) | | |
| Poorest | 1079 | 21.5 |
| Poorer | 1016 | 20.3 |
| Middle | 905 | 18.1 |
| Richer | 988 | 19.7 |
| Richest | 1023 | 20.4 |
| Birth order of the latest child | | |
| 1st child | 1915 | 38.2 |
| 2nd child | 1638 | 32.7 |
| 3 or more | 1458 | 29.1 |
| Place of residence | | |
| Urban | 1725 | 34.4 |
| Rural | 3286 | 65.6 |
| Region of residence (Division) | | |
| Barisal | 533 | 10.6 |
| Chattogram | 835 | 16.7 |
| Dhaka | 741 | 14.8 |
| Khulna | 524 | 10.5 |

(*Continued*)

**Table 2.** (Continued)

| Characteristics and categories | Number (n) | Percent (%) |
|---|---|---|
| Mymensingh | 603 | 12.0 |
| Rajshahi | 527 | 10.5 |
| Rangpur | 559 | 11.2 |
| Sylhet | 689 | 13.7 |
| Characteristics and categories | Number (n) | Percent (%) |
| Women's autonomy (decisions over own health) | | |
| Wife alone | 374 | 7.6 |
| Husband/wife together | 3236 | 65.4 |
| Husband alone/others | 1335 | 27.0 |
| Visited community clinics, temporary clinics or visited by family planning workers. | | |
| No | 2096 | 41.8 |
| Yes | 2915 | 58.2 |
| Covered by health insurance. | | |
| No | 5001 | 99.8 |
| Yes | 10 | .2 |
| MHS package utilization | | |
| Desirable/Ideal category | 1564 | 31.2 |
| Moderate | 2805 | 56.0 |
| Undesirable | 642 | 12.8 |

difference seen between the poorest and richest quintiles. More than 62% of the women who belonged to the richest household wealth index have utilized desired MHS package, and the rate for women with the poorest wealth quintile drops to 11.10%.

The desirable MHS package utilization also started to decrease with the child's birth order; the desirable MHS was reported to be higher when women were delivering their first child (Chi-square: 249.15, p<0.001). The desirable MHS use was also seen to be higher for women living in urban areas (Chi-square: 238.42, p<0.001) than their rural counterparts. Among eight divisions, the Khulna division had a higher percentage of desirable MHS package utilization than other divisions of the country. The women's autonomy had an insignificant association with MHS, although the percentage was lower when women could not make their own decisions (Chi-square:6.90, p< 0.141).

Table 4 represents the results derived from multinomial logistic regression analysis. The results reveal that women and their husbands' education significantly affected MHS package utilization when other variables were kept constant. Compared to the reference undesirable MHS category, women with higher education were 9.38 times (OR: 9.38, 95% CI: 4.30, 20.44) more likely to utilize the desirable MHS package than women with no education. Women and husbands' current working status did not affect significantly across groups; mother's age was also found to have varying degrees of association with the outcome variable.

Statistically, significant results were expectedly found with wealth index and MHS package; women with richest wealth quintile had 23.27 times (OR: 23.27, 95% CI: 12.69, 42.68) higher chance of desirable MHS utilization than their poorest counterpart. Women who belonged to the poorest wealth category were even 5.14 times (OR: 5.14, 95% CI: 2.13, 9.14) less likely to utilize even a moderate level of MHS package than their richest counterparts. The analysis further reveals that women having their 1st child had more than three folds (OR: 3.34, 95% CI: 2.39, 4.89) higher chance for desirable MHS than women having their 3rd or more child. Women living in urban areas had a better chance of utilizing desirable MHS than rural

**Table 3. Percentage distribution of women according to maternal health service package utilization.**

| Characteristics | Maternal health service (MHS) package utilization | | | Significance/Chi-square (p-value) |
| --- | --- | --- | --- | --- |
| | Desirable n (%) | Moderate n (%) | Undesirable n (%) | |
| Mother's age group | | | | |
| 15–19 years | 262 (30.1) | 513 (59.0) | 94 (10.8) | |
| 20–29 years | 970 (31.5) | 1735 (56.3) | 377 (12.2) | 16.72 (0.002) |
| 30–49 years | 332 (31.3) | 557 (52.5) | 171 (16.1) | |
| Marital status | | | | |
| Married | 1553 (31.4) | 2761 (55.8) | 631 (12.8) | 6.74 (0.036) |
| Divorced/widowed/separated | 11 (16.7) | 44 (66.7) | 11 (16.7) | |
| Religion | | | | |
| Islam | 1400 (30.5) | 2590 (56.5) | 598 (13.0) | 12.74 (0.002) |
| Others | 164 (38.8) | 215 (50.8) | 44 (10.4) | |
| Mother's highest level of education | | | | |
| Illiterate/No education | 25 (8.0) | 185 (59.3) | 102 (32.7) | |
| Primary School (Grade 5) | 206 (14.8) | 887 (63.8) | 298 (21.4) | 771.12 (0.000) |
| Secondary school | 789 (32.8) | 1385 (57.7) | 228 (9.5) | |
| Higher | 544 (60.0) | 348 (38.4) | 14 (1.6) | |
| Mother's current working status | | | | |
| Currently working | 485 (25.8) | 1095 (58.2) | 300 (16.0) | |
| Not working | 1079 (34.5) | 1710 (54.6) | 342 (10.9) | 54.25 (0.000) |
| Husband's highest level of education | | | | |
| Illiterate/No education | 83 (12.2) | 415 (61.2) | 180 (26.5) | |
| Primary School (Grade 5) | 309 (18.6) | 1048 (63.2) | 300 (18.1) | 784.84 (0.000) |
| Secondary school | 568 (34.7) | 934 (57.1) | 133 (8.1) | |
| Higher | 590 (61.3) | 356 (37.0) | 16 (1.7) | |
| Husband's occupation | | | | |
| Skilled manual | 548 (28.5) | 1122 (58.4) | 250 (13.0) | |
| Agricultural | 147 (16.0) | 572 (62.2) | 201 (21.8) | 392.62 (0.000) |
| Sales/service | 548 (34.7) | 872 (55.2) | 159 (10.1) | |
| Professional/Technical | 292 (63.5) | 157 (34.1) | 11 (2.4) | |
| Others | 29 (22.0) | 82 (62.1) | 21 (15.9) | |
| Household wealth (quintile) | | | | |
| Poorest | 120 (11.1) | 649 (60.1) | 310 (28.7) | |
| Poorer | 170 (16.7) | 672 (66.1) | 174 (17.1) | 1014.06 (0.000) |
| Middle | 260 (28.7) | 555 (61.3) | 90 (9.9) | |
| Richer | 379 (38.4) | 558 (56.5) | 51 (5.2) | |
| Richest | 635 (62.1) | 371 (36.3) | 17 (1.7) | |
| Birth order of the latest child | | | | |
| 1st child | 761 (39.7) | 1013 (52.9) | 141 (7.4) | 249.15 (0.000) |
| 2nd child | 502 (30.6) | 957 (58.4) | 179 (10.9) | |
| 3 or more | 301 (20.6) | 835 (57.3) | 322 (22.1) | |
| Place of residence | | | | |
| Urban | 763 (44.2) | 843 (48.9) | 119 (6.9) | 238.42 (0.000) |
| Rural | 801 (24.4) | 1962 (59.7) | 523 (15.9) | |
| Region of residence (Division) | | | | |
| Barisal | 129 (24.2) | 311 (58.3) | 93 (17.4) | |
| Chattogram | 233 (27.9) | 501 (60.0) | 101 (12.1) | |

*(Continued)*

**Table 3.** (Continued)

| Characteristics | Maternal health service (MHS) package utilization | | | Significance/Chi-square (p-value) |
|---|---|---|---|---|
| | Desirable n (%) | Moderate n (%) | Undesirable n (%) | |
| Dhaka | 281 (37.9) | 398 (53.7) | 62 (8.4) | 157.33 (0.000) |
| Khulna | 216 (41.2) | 284 (54.2) | 24 (4.6) | |
| Mymensingh | 153 (25.4) | 350 (58.0) | 100 (16.6) | |
| Rajshahi | 180 (34.2) | 294 (55.8) | 53 (10.1) | |
| Rangpur | 201 (36.0) | 291 (52.1) | 67 (12.0) | |
| Sylhet | 171 (24.8) | 376 (54.6) | 142 (20.6) | |
| Women's autonomy (decisions over own health) | | | | |
| Wife alone | 116 (31.0) | 213 (57.0) | 45 (12.0) | |
| Husband/wife together | 1054 (32.6) | 1775 (54.9) | 407 (12.6) | 6.90 (0.141) |
| Husband alone/others | 383 (28.7) | 773 (57.9) | 179 (13.4) | |
| Visited community clinics, temporary clinics or visited by family planning workers. | | | | |
| No | 693 (33.1) | 1150 (54.9) | 253 (12.1) | 6.29 (0.043) |
| Yes | 871 (29.9) | 1655 (56.8) | 389 (13.3) | |

women. Moreover, women living in the Khulna division had more than five times (OR: 5.84; 95% CI: 3.14, 10.06) higher chance of desirable MHS than women living in Sylhet, the reference division.

Table 5 provides discriminant analysis results. The analysis was conducted to assess whether the predictors in the model can distinguish those who utilized desirable from those who did not utilize the desirable MSH package. Wilks' lambda was significant, value: 0.78, chi-square: 1175.27, p<0.001, canonical correlation: .46, indicating that the model, which includes a number of predictors, significantly discriminate the two groups (desirable vs. undesirable). The tests of equality of group means provided that most variables except respondents' age, health decision, and division variables significantly predict or distinguish the two groups of MHS package utilization. The effect size of 0.22 represents the total variance associated with the discriminant function. The standardized function coefficient (Standardized Canonical Discriminant Function Coefficient) values indicate how each variable is weighted to maximize the discrimination of two groups. The results suggested that the wealth index contributed most (coefficient: 0.49) to distinguish those who utilized desirable MHS package from those who utilized undesirable package. The values from structure matrix correlation provide correlation values of predictors with the resulting discriminant function. The values suggested that the wealth index was very highly correlated (loaded) (correlation: 0.80) with the discriminant function that predicts who utilized desirable or undesirable MSH packages. Women's and husbands' education levels (correlation: 0.69 and 0.70 respectively) also loaded very highly on the discriminant function, although the weight was around 0.30 for these variables. The classification results indicate that the model correctly predicts 75.3% of the sample correctly; however, the prediction is much higher in the undesirable group than the desirable group.

## Discussion & conclusions

Guided by Andersen's behavioral model of health service use model, the study investigated the influence of predisposing, enabling, and need factors on maternal health service (MHS) package use. The component variables of MHS package use were important in the sense that one type of health service use facilitates the others. For instance, a study conducted in Noakhali,

**Table 4. Multinomial logistic regression model predicting the effect of predisposing, enabling, and health need factors on the MHS package utilization.**

| Characteristics | Maternal health service package utilization | | |
|---|---|---|---|
| | Desirable OR (95% CI) | Moderate | Undesirable (Reference) |
| Mother's age group | | | |
| 15–19 years (R) | 1 | 1 | |
| 20–29 years | 1.27 (0.91–1.82) | 1.17 (0.85–1.60) | |
| 30–49 years | 1.85 (1.16–2.93)* | 1.26 (0.84–1.88) | |
| Religion | | | |
| Islam | 1 | 1 | |
| Others | 1.48 (0.98–2.24) | 1.08 (0.75–1.56) | |
| Mother's highest level of education | | | |
| Illiterate/No education (R) | 1 | 1 | |
| Primary School (Grade 5) | 2.05 (1.22–3.44) * | 1.33 (0.98–1.80) | |
| Secondary school | 4.11 (2.44–6.94) * | 1.66 (1.19–2.31)* | |
| Higher | 9.38 (4.30–20.44) * | 3.08 (1.59–5.94)* | |
| Mother's current working status | | | |
| Currently working (R) | 1 | 1 | |
| Not working | 1.09 (0.86–1.38) | 1.016 (0.83–1.23) | |
| Husband's highest level of education | | | |
| Illiterate/No education | 1 | 1 | |
| Primary School (Grade 5) | 1.15 (0.81–1.62) | 1.05 (0.83–1.34) | |
| Secondary school | 2.07 (1.42–3.01) * | 1.41 (1.06–1.89)* | |
| Higher | 5.91 (3.07–11.35) * | 2.69 (1.47–4.93)* | |
| Husband's occupation | | | |
| Skilled manual | 1.84 (0.73–4.64) | 1.41 (0.65–3.02) | |
| Agricultural | 1.26 (0.49–3.24) | 1.22 (0.56–2.66) | |
| Sales/service | 2.34 (0.92–5.93) | 1.56 (0.72–3.38) | |
| Professional/Technical | 1.93 (0.62–6.01) | 1.24 (0.45–3.42) | |
| Others (R) | 1 | 1 | |
| Household wealth (quintile) | | | |
| Poorest (R) | 1 | 1 | |
| Poorer | 1.81 (1.31–2.50) * | 1.52 (1.21–1.98)* | |
| Middle | 3.94 (2.77–5.57) * | 2.05 (1.54–2.72)* | |
| Richer | 7.48 (4.96–11.23) * | 3.14 (2.22–4.49)* | |
| Richest | 23.27 (12.69–42.68)* | 5.14 (2.93–9.14)* | |
| Birth order of the latest child | | | |
| 1st child | 3.34 (2.31–4.89)* | 2.09 (1.53–2.85)* | |
| 2nd child | 1.74 (1.28–2.36)* | 1.58 (1.23–2.02)* | |
| 3 or more (R) | 1 | 1 | |
| Place of residence | | | |
| Urban | 1.65 (1.26–2.17) * | 1.27 (0.99–1.62) | |
| Rural (R) | 1 | 1 | |
| Characteristics | Desirable OR (95% CI) | Moderate | Undesirable |
| Region of residence (Division) | | | |
| Barisal | 1.21 (0.79–1.85) | 1.24 (0.89–173) | Reference category |
| Chattogram | 1.22 (0.83–1.78) | 1.44 (1.05–1.97)* | |
| Dhaka | 1.68 (1.11–2.58)* | 1.60 (1.12–2.29)* | |
| Khulna | 5.84 (3.41–10.06)* | 3.65 (2.24–5.94)* | |
| Mymensingh | 1.66 (1.10–2.51)* | 1.44 (1.04–2.00)* | |

*(Continued)*

**Table 4.** (Continued)

| Characteristics | Maternal health service package utilization | | |
| --- | --- | --- | --- |
| | Desirable OR (95% CI) | Moderate | Undesirable (Reference) |
| Rajshahi | 2.68 (1.71–4.20)* | 1.87 (1.27–2.74)* | |
| Rangpur | 3.88 (2.52–5.98)* | 1.94 (1.35–2.78)* | |
| Sylhet (R) | 1 | 1 | |
| Visited community clinics, temporary clinics or visited by family planning workers. | | | Reference category |
| No | 0.90 (0.72–1.12) | 0.97 (0.80–1.18) | |
| Yes (R) | 1 | 1 | |
| Model Fitting Criteria | -2 Log Likelihood: 7301; Chi-square: 1620.18; p: 0.000 | Pseudo R-square: Nagelkerke: 0.33 | Overall classification percentage: 63.2% |

R: Reference category

*: significant at <0.05 level.

Bangladesh, reported that women who did not receive ANC care were three times less likely to have facility delivery relative to women who received ANC [8]. The study revealed that education, household wealth, area of residence, and birth order significantly influenced the likelihood of using a desirable, undesirable, or moderate level of MHS package. The study found out that 12% of women utilize undesirable maternal healthcare packages who did not receive any kind of professional healthcare service, whether it was ANC or delivery care which was extreme in the context of utilizing or accessing services. Overall, around 68.2% of the women

**Table 5. Discriminant analysis: Maternal healthcare service package utilization discriminated by predisposing and enabling factors.**

| Predictive variables | Standardized function coefficients (Canonical discriminant function) | Correlations between variables and discriminant function (Structure matrix) | F (p-value) |
| --- | --- | --- | --- |
| Wealth index (Enabling factor) | 0.49 | 0.80[a] | 861.35 (<0.001) |
| Husband's level of education | 0.29 | 0.71[a] | 680.04 (<0.001) |
| Women's level of education | 0.30 | 0.69a | 634.74 (<0.001) |
| Place of residence | 0.18 | 0.40[a] | 219.74 (<0.001) |
| Birth order | 0.21 | 0.35 | 147.51 (<0.001) |
| Husband's occupation | -.07 | -0.30 | 126.99 (<0.001) |
| Mothers' current working status | -0.01 | 0.17 | 40.11 (<0.001) |
| Religion | 0.08 | 0.09 | 12.66 (<0.001) |
| Visited community clinics (Need factor) | -0.04 | 0.07 | 6.62 (<0.01) |
| Respondents' health decision | 0.05 | 0.05 | 3.67 (<0.055) |
| Respondent's age | 0.13 | -0.02 | 0.30 (0.58) |
| Division | -0.11 | -0.003 | .02 (0.98) |
| Discriminant analysis: Stepwise (Mahalanobis distance) | Classification results: **75.3%** of original grouped cases correctly classified. (Desirable: 47.5%; Undesirable: 88.1%) Prior probabilities (Compute from group sizes: Desirable: 0.31 Undesirable: 0.68 | Summary of canonical discriminant function: Box's M (Significance): 380. 84 (<0.001) Eigenvalue: 0.27 Canonical correlation: 0.46 Effect size: **0.22** Wilk's Lambda: 0.78 Chi-square value (df): 1175.27 (12) Significant: <0.001 | P-value derived from Test of equality of group mean, Criteria: Use of F-value: Entry: 3.84 Removal: 2.71 |

[a]Values that exhibit a loading of ±.40 or higher are considered as substantive. Discriminant loadings are reflection of the variance that the independent variables share with the discriminant function.

did not reach the desired level of healthcare package use, and the utilization was extremely low for the women who stand on the lowest socioeconomic spectrum. A retrospective study using data from the Health and Demographic Surveillance System (HDDS) in Matlab, Bangladesh, discovered that cesarean delivery increased with greater socioeconomic status, education, and utilization of antenatal care services [25]. Another study conducted in Bangladesh confirmed the significance of socio-demographic variables in the use of ANC and facility delivery [26].

The education level of women and their husbands significantly predicts the MHS package use and the utilization increases with the improvement of educational status. Relative to the undesirable category of MHS package use, the likelihood of receiving the desirable and moderate level of service package was 9.38 and 3.08 times higher for women who completed higher education compared to women with no formal education. This finding coincides with one study conducted in Uganda, which also employed a multinomial logistic regression technique for assessing the influence of predictors over the outcome. According to that study, women who completed a secondary or higher education level were more likely to utilize the desirable level of healthcare than their non-educated counterparts [21]. Another study that explored the utilization of ANC and health facility delivery care reported a strong contribution of women and partner's education on healthcare service utilization [8]. This provides strong evidence that education plays a vital role and improves maternal healthcare. It is imperative to improve the level of education in the community, not just women's education. Although women's autonomy on health decision-making did not significantly affect the MHS package use, the percentages of women utilizing desirable health packages were higher when women decided alone or with their husbands. A community-level focus on education at all levels might be helpful since several studies put attention on improving knowledge and literacy, which helps women and their families to take their health seriously and maintain care at every stage of pregnancy and childbirth [8, 21, 27–29]. Religion and the current working status of women and the type of husbands' occupation as parts of the social structure described in the HBM model also did not affect the service use significantly. However, it was seen that Muslim women were less likely to utilize the desired or moderate service package than women of other religions. Socio-cultural practices and religious beliefs may influence this decision [29]. Division of labor, heavy workloads during pregnancy, and very limited opportunities to ANC care can make pregnant women vulnerable, and the search for getting health service from health professionals get beyond the women's control at the time of delivery [30]; cultural practices can also affect negatively the child feeding practices which shows deep-rooted cultural stigmatization [31].

Household wealth, one of the important enabling factors of MHS package use, was found to have a greater predictive influence on service utilization which was one of the study's hypotheses. The difference between the poorest and richest groups certainly warrants a major issue requiring action. Relative to the undesirable category, women who belonged to the richest household had a more than 20 times higher chance of using the MHS package than their poorest counterparts. This huge difference in household wealth indicates healthcare inequality in the community. The discriminatory analysis confirmed that improving household wealth might increase the MHS package utilization since it has the strongest discriminatory function over desirable or undesirable MHS package utilization. One study conducted in Gabon focusing on the relationship of wealth status, health insurance, and maternal health service utilization reports that households' financial situation and enrollment in health insurance improve key maternal health service utilization [32].

Since according to the HBM model, enabling factors to have high mutability characteristics provide the means for use and improve the likelihood that use will occur [17]. The discrimination analysis in this study confirms household wealth as the most important discriminatory

variable. Since there was non-existent health insurance evident in the study population, improving household wealth as an enabling factor of service use should receive top priority for formulating policies or intervention of any kind [33]. The government of Bangladesh can also ponder about developing a health insurance policy, especially for marginalized families who are economically vulnerable and educationally deprived or underprivileged. A population-based study in Bangladesh also suggested that tailored healthcare services for poor people might help to reduce inequalities in service utilization [14].

The perceived or evaluated need factors in this study were considered to be how frequently women visited available community clinics, temporary clinics that were set up for targeting the decentralized rural population, and also whether any family welfare workers visited women in the last six months preceding the survey. The perceived need factor did not significantly affect the MHS package use much; however, a more specific and concrete measure of need factors might have helped to assess its effect on the overall utilization of MHS.

The discriminant analysis also reveals that the area of residence had a moderate level of correlation with discrimination function. Women living in urban areas better utilized the MHS than women living in rural areas, which entails that healthcare service utilization vary socially and geographically [34]. The reason could be women living in urban areas were better equipped to access health services, better transportation, or a socio-cultural context. As an administrative division, Khulna was in a better position among all other regions in the country where women utilized the MHS package better. The rural-urban and regional differences in health service utilization might be influenced by the various infrastructural differences of medical setup, service provision, health outcomes, socio-cultural gaps, the level of economic disparities, lack of resources, and opportunities [34, 35]. Another study conducted in India disclosed that urban women were more likely to use services than rural women due to low healthcare coverage, poor socioeconomic status, and poor exposure to education and mass media [36].

Moreover, the child's birth order was also found to have predictive capacity on MHS package utilization. A similar finding was evident in another study which entailed that the chances of using ANC and delivery service utilization decrease with the increased birth order [36]. The reason could be while women having their 1st child might feel insecure, in contrast, during higher-order births, women might feel confident due to her previous experience of delivery or other constraints that exacerbate the decision of health service utilization [37].

The study's choice of predictors and the process of developing indicator outcome variables were not devoid of limitations. The variables selected to be the components of the MHS package cannot be declared comprehensive for overall maternal healthcare service utilization. Since there are other factors like components of ANC care, its effective utilization, patient-providers quality of communication, waiting time, satisfaction level of the patients, etc., were not included in the service package. The predictors were chosen based on Andersen's behavioral model, which has its own limitations on explaining the overall healthcare use. For instance, the model has been criticized for not considering cultural norms, social networks, and interaction to be included; however, the expanded model did try to consider the race and ethnicity variables' effect on the use of services. The predisposing factors of the HBM talked about health beliefs which include knowledge, attitudes, and values, were not considered for the study, although women's education and their decision-making capabilities on their own healthcare were included in the study as proxy measures. Moreover, the secondary nature of the data collected through retrospective interviews may have recall bias, inaccurate information, and omission of important details. And the cross-sectional nature of the data also hinders the development of causal (cause-effect) relationships among variables. The BDHS data also did not offer any program-specific information through which it might have been possible to differentiate which specific regions have better statistics.

Furthermore, the HBM model only discussed the factors that could affect health service use; however, policy options for changing predictor status, which does not have high mutable characteristics, might take years to positively influence the outcome. And to improve the household wealth that enables and facilitates health service use might also be related to other predisposing factors of social structure like education or occupation with low mutability characteristics. Therefore, parallel investments in programs which targets poverty alleviation and increasing family income, wealth, and security, universal school education program [33] and reducing urban-rural or regional healthcare access [34, 36] gap might bring effective change in maternal health care service utilization. Moreover, the government can also strengthen the Maternal and Neonatal Health Initiative that was started in selected districts of Bangladesh with the aim to reduce inequitable utilization of health services. The initiative was reported to have improved health service use and lowered inequality across socioeconomic groups [38].

Furthermore, the cultural factors, health beliefs, attitudes, and values should also be considered while developing any policies and interventions to change the service use. Women of disadvantaged communities regarding household wealth, education, or geography need to be prioritized for equitable health access. Health-related policies that address socioeconomic disparities and inequities and generate public awareness to access and utilize available healthcare services might reduce high maternal mortality in Bangladesh.

## Supporting information

**S1 File.**
(RAR)

## Author Contributions

**Conceptualization:** Md. Ruhul Kabir.

**Data curation:** Md. Ruhul Kabir.

**Formal analysis:** Md. Ruhul Kabir.

**Methodology:** Md. Ruhul Kabir.

**Resources:** Md. Ruhul Kabir.

**Software:** Md. Ruhul Kabir.

**Supervision:** Md. Ruhul Kabir.

**Validation:** Md. Ruhul Kabir.

**Visualization:** Md. Ruhul Kabir.

**Writing – original draft:** Md. Ruhul Kabir.

**Writing – review & editing:** Md. Ruhul Kabir.

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
