## [Decision Letter · Decision Letter 0]

9 Sep 2021

PONE-D-21-21996Adopting Andersen’s behavior model of health service use to identify factors influencing maternal healthcare service utilization in BangladeshPLOS ONE

Dear Dr. Kabir,

Thank you for submitting your manuscript to PLOS ONE. After careful consideration, we feel that it has merit but does not fully meet PLOS ONE’s publication criteria as it currently stands. Therefore, we invite you to submit a revised version of the manuscript that addresses the points raised during the review process.

We look forward to receiving your revised manuscript.

Kind regards,

Russell Kabir, PhD

Academic Editor

PLOS ONE

Journal Requirements:

2. Please ensure that you refer to Figure 1 & 2 in your text as, if accepted, production will need this reference to link the reader to the figure.

Reviewers' comments:

Reviewer's Responses to Questions

**Comments to the Author**

1. Is the manuscript technically sound, and do the data support the conclusions?

Reviewer #1: Yes

Reviewer #2: Partly

Reviewer #3: Yes

2. Has the statistical analysis been performed appropriately and rigorously? 

Reviewer #1: Yes

Reviewer #2: Yes

Reviewer #3: Yes

3. Have the authors made all data underlying the findings in their manuscript fully available?

Reviewer #1: Yes

Reviewer #2: Yes

Reviewer #3: Yes

4. Is the manuscript presented in an intelligible fashion and written in standard English?

Reviewer #1: Yes

Reviewer #2: No

Reviewer #3: Yes

5. Review Comments to the Author

Reviewer #1: 1. Please omit Factors; Bangladesh from keyword.

2. write abbreviated words for MHS in abstract.

3. Please avoid repetitive words.

4. Andersen, R. M. (1995).....written twice in Reference section. Recheck references

Reviewer #2: Title- Extra word used in title. It needs to be more scientific/ research title. Type of research method need to be mentioned

Abstract- Objective is not addressed properly as mentioned in the title. The method is not completely mentioned. The process of data collection and analysis is not mentioned. The grammar/ tense need to be correct in result section. What is Andersen's model need to be mentioned? Factors are not addressed properly in result section.

Introduction- Andersen's model related information need to be added. Starting sentence/ para is not attractive. The problem is not addressed adequately.

Method- Method section need to be concise. The ethical approval, conflict of interest, limitation of the study is not mentioned. The data extraction process need to be clearly mentioned.

Result- Section is OK, but as mentioned table in BDHS report is similar in this manuscript, then need to be changed.

Discussion- Discussion section is large enough, need to be concise. First para should contain the major findings of this research. How other research similar or dissimilar with this findings need to be mentioned properly.

Reviewer #3: The manuscript is well written, technically sound and backed by analysis of data used from the secondary source BDHS which is available. The author has concluded mentioning the broader issues to address such as poverty, geographical disparity and in health services. However, there is a scope to further examine the influence of having maternal health scheme or health protection schemes which are currently being tested in few districts of Bangladesh and that could give a strategic direction to the policy makers to find the desirable use of MHS in Bangladesh.

The methodology was well articulated and the adaptation of HBM with categorization of accessibility issues of MHS services has been illustrated adequately. The sampling was done following standard process and statistical analysis was done with rigor. The tables and analysis have been helpful to understand the logic and inference drawn by author. Regarding data availability and all the analysis, the authors has ensured that all data underlying the findings described in their manuscript are fully available without restriction. To my knowledge and understanding the author has written the manuscript in an intelligible fashion and the article is clear and correct without any ambiguity.

As an additional comments to author, author could review literature available specifically icddr,b which has many publications in relation the MNCH as they have been doing longitudinal study using their Matlab database. The other important issue that author could consider during sampling, one or two district's BDHS data could be purposively considered for analysis by using the adapted HBM framework to have a comparative picture of factors that influences which the author mentioned in his conclusion. The purposive sampling of these special districts are currently having government's (MOHFW and Health Economic Department) maternal health scheme and health protection scheme and this is done to test to address equity issues in health especially in maternal health. The government's schemes will be adapted to achieve UHC in near future. Therefore, this was missed opportunity that could have been considered or may be in future studies to help policy makers and planners. Apart from this I do not have any concerns or specific comments.

6. PLOS authors have the option to publish the peer review history of their article (what does this mean?). If published, this will include your full peer review and any attached files.

Reviewer #1: **Yes: **Farzana Ahmed

Reviewer #2: **Yes: **Dr. Abu Sayeed Md. Abdullah

Reviewer #3: **Yes: **Dr. Munir Ahmed

---

## [Author Response · Author response to Decision Letter 0]

26 Oct 2021

Adopting Andersen’s behavior model to identify factors influencing maternal healthcare service utilization in Bangladesh

Author’s response (overall): I would like to show my gratitude to you all (Dear Editor and three reviewers) for your kind response and valuable comments, which have helped me improve my work. I took a lot of time to wonder how I could improve this work and go forward with your comments effectively. I tried to incorporate your ideas, suggestions, and corrections into my manuscript, and I, again, thank you for your kind response. I have used blue color, where I have made major changes; however, I have also paraphrased some sentences and deleted some. 

Reviewer #1: 1. Please omit Factors; Bangladesh from keyword.

Author's response: The keywords are omitted.

2. write abbreviated words for MHS in abstract.

Author's response: Full meaning of MHS is already mention in Method section.

3. Please avoid repetitive words.

Author's response: I’ve gone through the manuscript and tried to avoid repetitive words.

4. Andersen, R. M. (1995).....written twice in Reference section. Recheck references.

Author's response: The duplicate reference is deleted.

Reviewer 2: 

This is a good research work. However I have mentioned below very few recommendation

Title- Extra word used in title. It needs to be more scientific/ research title. Type of research method need to be mentioned.

Author’s response: I’m proposing the following titles for your review because I am unsure what’s to change. I am dropping health service use from the title, although I still prefer the previous one.

1. Adopting Andersen’s behavior model to investigate factors affecting maternal healthcare service use in Bangladesh.

2. Identification of factors that influence maternal health service use in Bangladesh by applying Andersen’s behavioral model of healthcare utilization.

3. Application of Andersen’s behavior model to analyze factors influencing maternal healthcare service use in Bangladesh.

4. Analysis of influencing factors affecting maternal health service use in Bangladesh: An application of Andersen’s behavior model

Abstract- Objective is not addressed properly as mentioned in the title. The method is not completely mentioned. The process of data collection and analysis is not mentioned. The grammar/ tense need to be correct in result section. What is Andersen's model need to be mentioned? Factors are not addressed properly in result section. 

Author’s response: The limitation of 250 words restricted the opportunity to disclose every important aspect in the abstract. I tried to accommodate which I felt necessary to be included in the abstract. I mentioned Andersen’s model because the theoretical framework was adapted from this model; hence, I thought it’s important for the readers to know what’s the paper is based on. I’ve mentioned most of the factors that consistently affect MHS use. I’ve tried to rewrite the abstract by following your suggestions.

Introduction- Andersen's model related information need to be added. Starting sentence/ para is not attractive. The problem is not addressed adequately.

Author’s response: I’ve included Andersen’s model in the last paragraph of the introduction section and discussed adequately in the methods section. I’ve tried to improve the starting paragraph of the introduction and the problem statement as you mentioned.

Method- Method section need to be concise. The ethical approval, conflict of interest, limitation of the study is not mentioned. The data extraction process need to be clearly mentioned. 

Author’s response: Thank you for your suggestions. I’ve tried to concise the method section. The theoretical framework development made the method section a bit long. The ethical statement (although this is a secondary analysis) and conflict statement are included as acknowledgments. The data extraction process is mentioned in the data source section (method). Limitations of the study are also mentioned in the last section of the discussion.

Result- Section is OK, but as mentioned table in BDHS report is similar in this manuscript, then need to be changed. 

Author’s response: Thank you.

Discussion- Discussion section is large enough, need to be concise. First para should contain the major findings of this research. How other research similar or dissimilar with this findings need to be mentioned properly.

Author’s response: I’ve gone through the discussion session and tried to incorporate the suggestion you’ve provided. Discussion and conclusion were written altogether, that’s why it might seem a bit long.

Reviewer 3:

The manuscript is well written, technically sound and backed by analysis of data used from the secondary source BDHS which is available. The author has concluded mentioning the broader issues to address such as poverty, geographical disparity and in health services. However, if there was a scope to further examine the influence of having maternal health scheme or health protection schemes which are currently being tested in few districts of Bangladesh and that could give a strategic direction to the policy makers to find the desirable use of MHS in Bangladesh.

The methodology was well articulated and the adaptation of HBM with categorization of accessibility issues of MHS services has been illustrated adequately. The sampling was done following standard process and statistical analysis was done with rigor. The tables and analysis have been helpful to understand the logic and inference drawn by author. Regarding data availability and all the analysis, the authors has ensured that all data underlying the findings described in their manuscript are fully available without restriction. To my knowledge and understanding the author has written the manuscript in an intelligible fashion and the article is clear and correct without any ambiguity. 

As an additional comments to author, author could review literature available specifically icddr,b which has many publications in relation the MNCH services ( reference) as they have been doing longitudinal study using their Matlab database. The other important issue that author could consider during sampling, one or two district's BDHS data could be purposively considered for analysis by using the adapted HBM framework to have a comparative picture of factors that influences which the author mentioned in his conclusion. The purposive sampling of these special districts is currently having government's (MOHFW and Health Economic Department) maternal health scheme and health protection scheme, and this is done to test to address equity issues in health especially in maternal health. The government's schemes will be adapted to achieve UHC in near future. Therefore, this was missed opportunity that could have been considered or may be in future studies to help policy makers and planners. Apart from this I do not have any concerns or specific comments. 

Author’s response: 

Thank you very much for your thought-provoking input in my work. Of course, there is room for further studies on this topic. However, as you know, BDHS data do not have any district-specific information; hence, it is not possible to break down further. I’ve included the division-specific breakdown and the rural vs. urban differences in health service use. I know about the longitudinal study and surveillance system icddr,b has set up in Matlab with its development partners to test different interventions to improve the maternal and child health situation. I’m yet to know the government plan on maternal health scheme and how comprehensive it will be to protect and address issues that have been evident in different studies. I tried to incorporate some of the literature which I deemed relevant to this study. However, the national statistics are different, and it would have been great if I could recommend any specific scheme to policymakers from the data I analyzed. There is a lot of buzz around this maternal and child health issue, and it’s about time government implements comprehensive nationwide measures to change the scenario. I am considering future studies on the issue you’ve mention; however, it will also depend on the data availability.

---

## [Editor Report · Decision Letter 1]

11 Nov 2021

Adopting Andersen’s behavior model to identify factors influencing maternal healthcare service utilization in Bangladesh

PONE-D-21-21996R1

Dear Dr.Kabir,

We’re pleased to inform you that your manuscript has been judged scientifically suitable for publication and will be formally accepted for publication once it meets all outstanding technical requirements.

Kind regards,

Russell Kabir, PhD

Academic Editor

PLOS ONE